# Four Months of Exercise Intervention Improved Visuomotor, Functional and Cardiorespiratory Capacity in a Patient with Metastatic Uveal Melanoma

**DOI:** 10.3390/reports8040260

**Published:** 2025-12-10

**Authors:** Michael Mendes Wefelnberg, Stefanie Hennigfeld, Michael Simon, Philomena Wawer Matos, Ludwig M. Heindl, Alexander C. Rokohl, Paul Bröckelmann, Freerk T. Baumann

**Affiliations:** 1Department 1 of Internal Medicine, Center for Integrated Oncology Aachen, University Hospital Cologne, 50937 Cologne, Germany; 2Dynamic Eye, Institute for Visual and Cognitive Training, 50937 Cologne, Germany; 3Department of Ophthalmology, Faculty of Medicine and University Hospital Cologne, University of Cologne, 50937 Cologne, Germany

**Keywords:** uveal melanoma, exercise training, visuomotor training, Cyberknife, immunotherapy, rehabilitation, supportive care

## Abstract

**Background and Clinical Significance**: Uveal melanoma (UM) is an ocular malignancy with high mortality for which supportive therapies to mitigate disease and treatment-related side effects are lacking. Exercise therapy is one of the most versatile symptom-management strategies in oncology. We investigated the effects of a 4-month combined exercise intervention to restore and stabilize disease and treatment-related side effects. **Case Presentation**: A moderately active 61-year-old woman, diagnosed with metastatic UM in the right eye and treated with Cyberknife radiation, presented with diminished visual motor capacity due to disease-related loss of stereopsis and visual field reduction, without systemic comorbidities. The main outcome measures were visuomotor and functional tests (VFT) and cardio-pulmonary exercise testing (CEPT). All functional and most visuomotor tests demonstrated meaningful improvements between baseline and post-intervention by 7–128% and 8–24%, respectively. The CPET-derived parameters (% for $V_{\dot{E}}$, $V{\dot{O}}_{2}$, PPO, CEPT duration) showed improvements between 10 and 30% throughout the 4-month period. **Conclusions**: Data from this case report indicate that the 4-month exercise intervention yielded a consistent pattern of improvement in most VFT dimensions and cardio-pulmonary capacity. Interestingly, our data imply that post-radiation declines in visuomotor capacity recovered and expanded with enhanced manual dexterity. Future investigations need to extend our findings to a larger cohort of UM patients.

## 1. Introduction and Clinical Significance

Uveal melanoma (UM) is the most common intra-ocular malignancy globally, with an incidence of six to eight cases per million people and high overall mortality due to aggressive metastatic spread [1,2]. Metastases occur in 40 to 50% of patients, predominantly in the liver, and are associated with poor prognosis and a low overall survival of 6–12 months despite treatment [3]. On top of the aggressive metastatic nature of the disease, patients frequently suffer from visual impairments, mostly related to binocular fusion and visual field, induced by disease and medical treatment, surgery, or radiation, that can lead to severe functional declines. These declines encompass climbing stairs, pouring drinks, navigating in public gatherings and driving. The consequences often are social isolation, anxiety, depression and an overall declining health-related quality of life [4,5]. An observational study from the UK monitoring patients for 10 years after treatment demonstrated continuous declines in most of these variables in up to 70% of patients. Thus far, there have been no established supportive therapies or rehabilitation strategies of any kind to mitigate symptoms and side effects and improve people’s lives. Previous investigations emphasized the potential of targeted exercise interventions to improve overall eye health and balance in people with ocular impairments [6,7]. Furthermore, experimental studies imply the potential of improved manual dexterity to compensate for visuomotor deficiencies in eye–hand coordination (EHC) performance [8,9].

Cyberknife, a stereotactic radiosurgery method, is a very precise and relatively novel form of radiation therapy that minimizes damage to surrounding tumor tissue and thus preserves the eye globe. Serious side effects and complications can occur nonetheless, such as secondary glaucoma, lens opacities and radiation-induced retinopathy [10]. These side effects can lead to deteriorated visual acuity or even blindness [11].

Metastatic disease in UM is very aggressive [12]. Thus far, there is no standard to treat metastatic disease in UM [13]. Recently, immune checkpoint inhibitors, more precisely, a combination of nivolumab and ipilimumab, have demonstrated some promising results, with an overall response rate of up to 18%. However, this combination yields increased rates of immune-related adverse reactions in up to 60% of patients. Among the most frequent adverse reactions are diarrhea (60%), liver toxicity reflected in increased aspartate aminotransferase (AST) (49%) and alanine aminotransferase (ALT) (40%) levels, pruritus (40%), hypothyroidism (37%) and rash (31%) [14]. In a pilot study of 10 patients with stage IV melanoma under immunotherapy, telehealth home-based exercise improved different quality-of-life dimensions, substantially enhanced physical activity behavior and reduced sedentary behavior [15].

Therefore, the primary aim of this case report was to investigate the effects of a 4-month exercise intervention (a combination of visual-coordinative training + HIIT) post-radiation and under immunotherapy on visuomotor and functional tests (VFT) and cardiorespiratory capacity in a moderately active middle-aged woman diagnosed with uveal melanoma. We postulated that the integrated intervention has the capacity to enhance visuomotor function and cardiorespiratory fitness and consequently contribute to an amelioration in physical activity levels, overall quality of life and ultimately, a reduction in treatment-related adverse effects.

## 2. Case Presentation

### 2.1. Patient

A 61-year-old moderately active woman was diagnosed with metastatic uveal melanoma with ciliary body and iris involvement in the right eye (ICD-10 code: C69.4). She had a history of malignant melanoma on the right leg in her 40s. In late August 2022, she was diagnosed with uveal melanoma. An MRI scan in September 2022 revealed seven suspected metastatic liver lesions, which were histologically confirmed through a liver biopsy in November 2022 (HLA A02:01 negative; BRAF wild type; GNAQ mutated). In April 2023, progression of liver metastasis (n > 20) was detected via MRI scan. The primary tumor originated from the ciliary body, showed iris involvement at 2 o’clock and extended into the macula at 1–4 o’clock. The total size was 17 mm with a 9.7 mm prominence (as visualized by fundus photography, see Figure A1) and an accompanying amotio inferior. Otherwise, the retina was mid-peripheral adjacent. The optic disk was marginally sharp and vital. In the left eye, she had severe epiretinal membrane formation (epiretinal gliosis). At the time of study participation, the patient used eyeglasses and was free of known circulatory, respiratory or any other relevant deficits or comorbidities. At the time of study inclusion, apart from reduced visual acuity, the patient was free of any cancer-related impairments. Table 1 lists all the relevant biometrical patient characteristics.

### 2.2. Design

This is an interventional, prospective case study of a patient with metastatic uveal melanoma under acute medical treatment who volunteered to participate in a targeted exercise intervention accounting for cardiovascular and functional deficits related to visual impairment from eye cancer [4,5]. Before participation, she was informed of the risks and stresses associated with the protocol and gave informed consent for the test and publication of her results. The investigation was conducted in accordance with the standards of the Declaration of Helsinki, approved by the Ethics Committee of the University Hospital of Cologne and registered at the German Clinical Trial Register (DRKS) within the scope of the EyeCanMoveiT-Study (No. 22-1062_1; Clinical Trial Register ID: DRKS00031207).

### 2.3. Therapy and Treatment

The patient received Cyberknife radiation therapy for the primary tumor in November 2022 (stereotactic irradiation with 1 × 21 Gy, corresponding to a 70% isodose). For the treatment of liver metastases, four cycles of immune checkpoint inhibitor therapy were applied from December 2022 to March 2023 (each consisting of Ipilimumab 3 mg/kg body weight and Nivolumab 1 mg/kg body weight). She participated in the exercise program during the entire treatment phase.

### 2.4. Exercise Intervention

The exercise therapy program was implemented in a clinical setting at the Center for Integrated Oncology, University Hospital Cologne, Germany, from 12 November 2022 to 2 March 2023. To compensate for the treatment-induced visual capacity decline and to stimulate the microvascular, retinal network to prevent radiation retinopathy, the intervention consisted of two components: visual-coordinative exercises and HIIT. Prior to commencing first-line treatment (Cyberknife radiation), the patient underwent two preparatory sessions spaced 48 h apart to familiarize herself with the program. The training regimen consisted of two weekly exercise sessions, lasting 60 min each. These sessions were divided into 30 min of HIIT and 30 min of visual-coordinative exercises. HIIT was performed on a cycle ergometer. It included a 5 min warm-up and cool-down, along with 20 min (5 cycles) of high-intensity cycling (≥90% of $V{\dot{O}}_{2\max}$) and recovery periods (at 60% of $V{\dot{O}}_{2\max}$), maintaining a 1:2 ratio [16]. The visual-coordinative exercises, developed in collaboration with experienced visual trainers, were designed to enhance coordination and balance in both the upper and lower extremities. For balance, the patient started with fundamental exercises such as line walking, standing on various surfaces and wobble boards. Once a high level of proficiency was achieved, difficulty progressed to more challenging one-leg tapping exercises with static and dynamic visual cues.

The eye–hand coordination training involved fine and gross motor tasks like tapping, throwing, threading and pouring. Visual stimuli, including static images providing task cues, and dynamic stimuli, such as changing colors on a tablet app presented frontally or laterally, were incorporated to add complexity. The fundamental elements of the exercise program are presented in Table A1.

### 2.5. Measurement and Assessments

Over the course of the study, the patient’s data were collected on five separate occasions: baseline, prior to Cyberknife radiation (T0), post-Cyberknife radiation (T1), prior to immunotherapy (T2), intermediate (T3) and post-intervention/immunotherapy (T4). Prior to each data collection, the participant was asked to avoid any vigorous exercise, caffeine and alcohol intake for at least 24 h. After the first assessment, the patient participated in a brief familiarization phase prior to Cyberknife radiation treatment to get to know the exercise protocol. The second data collection took place immediately after radiation treatment. From then on, assessments were conducted roughly every four to five weeks, accounting for immunotherapy treatment protocol. Briefly, the assessment protocol consisted of CPET, VFT, daily physical activity (PA) measurement and patient-reported outcomes (PRO). On the second occasion, exclusively VFT and PA were undertaken to detect radiation-induced acute functional and visuomotor impairments as well as impacts on PA behavior. Generally, PA measurements took place from T0 and T3. At T2 some of the VFT could not be conducted due to sudden vertigo and nausea. For graphical representation, missing values were interpolated using the average of T1 and T3 values. See Figure 1 for a detailed illustration of the study timeline. Complementing the assessment protocol, pre- and post-intervention microvascular data and blood work data over the interventional period and beyond were collected.

#### 2.5.1. Cardio-Pulmonary Testing

Pulmonary gas exchange was assessed on a breath-by-breath basis using a stationary metabolic chart (Cortex, Leipzig, Germany ) synchronized with a cycle ergometer (Ergoline 900, Hamburg, Germany). The metabolic chart underwent calibration before each laboratory visit, following the manufacturer’s guidelines, using a gas mixture of known composition and ambient air. The patient was positioned on a bed and equipped with a silicone mask (Hans Rudolph, Kansas City, MO, USA) connected to a metabolic analyzer, as well as a chest-worn HR belt (Polar H10, Oyo, Finland). To outline the procedure briefly, the patient engaged in an incremental ramp test (30/15 protocol) on the cycle-ergometer, continuing until she reached task failure. This test aimed to determine various parameters, including peak ${\dot{V}O}_{2}$, peak power output (PO) achieved at peak ${\dot{V}O}_{2}$ and maximum heart rate (HR max). Task failure was defined as occurring when the cycling cadence dropped below 70 revolutions per minute for more than 10 consecutive seconds, despite strong verbal encouragement from the research team.

#### 2.5.2. Visual and Functional Testing

The VFT consisted of seven different assessments to evaluate various aspects of visual, visuomotor and functional performance. We utilized the Computerized Binocular Home Eye Exercise System/Binocular Vision Assessment, Version 3.02 (BVA) (HTS, Gold Canyon, AZ, USA) to assess binocular fusion capacity. The BVA was performed on a computer with a distance of 40 cm wearing bicolor glasses (red and green). To achieve three-dimensional vision, with increasing difficulty, two flat squares had to be merged into a small square, the position of which had to be confirmed by pressing the arrow keys accordingly. The test measures the limits of fusion capability, also referred to as fusion width. It measures the maximum deposition angle in prism diopters (pdpt) between the two eye-separated images that can still be fused into one image by vergence movements. A distinction is made between the fusional vergence of distance and proximity. The value Bi (base-in) comprises the divergent vergence range (negative vergence) and BO (base-out) the convergent vergence range (positive vergence). In addition, so-called recovery values were determined both at distance and near (BIRec and BORec). With each keystroke, the large square moved apart, gradually forming two squares that became increasingly challenging to fuse.

For balance assessment, the One-Leg Stance Test (OLS) was applied. Participants underwent three trials of the OLS, both with eyes closed and open. Further details regarding the test procedure can be found elsewhere [17]. For gait speed assessment, we used the well-established Timed Up and Go Test (TuG). This test required the patient to rise from a chair upon command, walk a distance of three meters, turn around and return to the chair as quickly as possible to measure their gait speed. Functional capacity assessment was conducted via a 30 s sit-to-stand test (30 s StS). Here, the patient was tasked with standing up and sitting down as many times as possible within a 30 s time frame to evaluate her functional capacity. To assess EHC we utilized an established manual dexterity test, the Purdue Pegboard Test (PPT) (Lafayette Instruments, Lafayette, IN, USA), that demands three-dimensional visual capacity and a central and a peripheral visuomotor reaction time test using the D2 Visuomotor Training System (D2) (Dynavision, West Chester, OH, USA). In the PPT, the patient was required to place as many pins as possible in a vertical column of holes on a board within 30 s. The test included three rounds for the right hand, left hand, and both hands. The D2 device features a vertical panel with 64 light-emitting buttons arranged in concentric circles. For central visuomotor reaction time (CVRT), we measured the time it took the patient to identify a stimulus (a lit-up dot) by moving from the “home” dot (sensory reaction time) and initiating a reaction by tapping on the lit-up dot (motor reaction time). For peripheral visuomotor reaction (PVRT), we employed a test that demanded the patient to tap as quickly as possible on randomly lit-up buttons, one at a time, within one minute while maintaining focus on the center of the board. Further details about the test procedures are described elsewhere [18].

#### 2.5.3. Routine Ophthalmological Examinations and Laboratory Data

The patient underwent detailed prospective ophthalmological examinations at the Department of Ophthalmo-Oncology at the University Hospital Cologne. Among these were best-corrected distant visual acuity using an automatic refractometer (ARK-1s, Nidek, Tokyo, Japan), inner ocular pressure (IOP) (ic100, Icare, Vanda, Finnland) and fundus examination (OPTOS, P200 DTx, Dunfermline, Scotland, UK) to determine tumor location and size. Additionally, retinal blood flow and vessel diameters were analyzed via optical coherence tomography angiography (OCTA) (Optovue Solix, Visionix, Jerusalem, Israel). For OCTA imaging the patient was instructed to abstain from moderate to vigorous physical activity, caffeine, nicotine and alcohol consumption 24 prior to imaging. The OCTA images were taken within 10 days before and after T0 and T4 by an experienced operator under standardized mesopic lighting conditions in late October 2022 and mid-March 2023, respectively. On each occasion, one image of the central macula region (6.4 × 6.4 mm) of the tumor eye (right eye) was taken. In the case of low scan quality (<6), imaging was repeated until a satisfactory quality was achieved. The retrieved parameters, automatically calculated by the proprietary software of the OCTA device (AngioAnalytics, version 2018.0.0.18), comprised foveal avascular zone (FAZ) area, FAZ perimeter, macular flow density (FD) and vessel density (VD) of the superficial and deep plexus. Laboratory data were retrieved from the hospital information system (ORBIS, Dedalus Healthcare Group, Florence, Italy) to monitor hematocrit levels and liver function blood markers.

#### 2.5.4. Patient-Reported Outcomes

The patient outcomes (PROs) comprised the European Organisation for Research and Treatment of Cancer (EORTC) questionnaires EORTC_C30 along with the complementary module EORTC_OPT30, specifically designed for eye cancer patients, and the Hospital Anxiety and Depression Scale (HADS). Both of these surveys assess various dimensions of quality of life (QoL), while the OPT30 module has a particular focus on evaluating visual impairments. Additionally, after the onset of immunotherapy, the patient was asked for prototypical adverse effects reported in the literature [14].

#### 2.5.5. Physical Activity Behavior

To assess PA behavior, we applied the move IV accelerometer and the accompanying software, DataAnalyzer, (movisens GmbH, Karlsruhe, Germany) to retrieve and analyze collected data. The accelerometer consists of a three-axial acceleration sensor, which can be attached at different locations and allows measurements for up to 7 days. The patient was instructed to wear the sensor for 7 days over her left hip at the level of the iliac crest using the belt that came with the device. It is noteworthy that the chosen location cannot adequately capture cycling activity. Upon data retrieval, the output-sampling rate was set to 60 s. In addition to step count and activity behavior, the software estimates energy expenditure and metabolic equivalent (MET).

### 2.6. Outcomes

The patient characteristics are provided in Table 1. Briefly summarized, the patient had a BMI of 20.68, an inner ocular pressure within the normal range, a visual acuity that was reduced by ~40% in the right eye and 60% in the left eye and an impaired but still functioning stereopsis derived by the BVA. Eye, hand and foot dominance were on the right side. Meaningful changes over the course of the study (from T0 to T4) were observed for all cardiorespiratory and metabolic variables, for all functional tests and in most dimensions of the visual and visuomotor tests. In detail, the CPET-derived parameters (% for $V_{\dot{E}}$, $V{\dot{O}}_{2}$, PPO and CEPT duration) showed a steady improvement between T0 and T3 in response to 17 exercise sessions (+2 familiarization sessions) of 15–51% and a decline between T3 and T4 of roughly 4–14%. Over the entire course of the study (T0–T4), improvements for $V_{\dot{E}}$, $V{\dot{O}}_{2}$, PPO and CEPT duration were between 10 and 30. Overall, the patient’s condition deteriorated during the last third of the intervention period, after two months of immunotherapy. The patient maintained hemoglobin concentrations within reference values throughout the 4-month exercise period (see Table 1).

All functional tests demonstrated meaningful improvements between T0 and T4 by 7–128% without notable declines after radiation (T1). The BVA data demonstrated low binocular fusion capacity at T0, unchanged values after radiation (T1), large improvements at T3 and loss of binocular fusion capacity at T4. For the visuomotor tests, results were heterogeneous. While EHC and CVRT exhibited improvements of 8–24%, PVRT showed a deterioration of 14% between T0 and T4. In detail, CVRT values deteriorated by 11% in the sensory dimension but were enhanced by 36% in the motor dimension, accumulating to an overall improvement of ~24%. VFT changes over the course of the study are illustrated via z-transformed scores in Figure 2. All VFT results are provided in Table 1.

Comparison of the OCTA data pre- and post-intervention showed a stable FAZ, reduced FAZ perimetry (by ~7%) and increased FD (by~2%). VD depicted a pattern of decrease in the superior hemisphere (by ~6% superficial and ~14% deep) and increase in the inferior hemisphere (by ~2% superficial and ~10% deep). VD in the whole image decreased in the deep and superficial plexus, respectively (~2% each). OCTA data are listed in Table A2. Pre- and post-intervention images of the vascular plexi and fovea cross-section, including blood flow visualization, are provided in Figure A1.

For PA results (step count, energy expenditure and active time) we observed a pattern of reduction post-radiation (T1) by ~2–42%, followed by a large increase between T1 and T3 by ~5–45%. Analogically, inactive time showed an increase and decrease pattern by ~16, −5, and 10%, respectively. Overall (T0–T3), decreases of ~2–16% (step count, energy expenditure and active time) and an increase of ~10% for inactive time were obtained. PA data are listed in Table 2.

For most PRO readings, no overall trend could be observed. Most remarkably, global health status and some vision- and visual-function-related scores demonstrated deteriorations between T0 and T4. Other scores demonstrated a variety of developments. The HADS values indicate mild depressive and moderate anxiety symptoms at T0. Throughout the intervention anxiety levels decreased to mild but increased dramatically to severe at T4. Depression scores remained stable. At the onset of immunotherapy, the patient reported a variety of continuously exacerbating prototypical adverse reactions, including rash, itchy skin, headache, dry mouth and, after the second immunotherapy cycle, loss of power and faster exhaustibility as well as unclassifiable pain. Concerning liver blood markers, AST and ALT remained generally stable and within referent values throughout the exercise intervention period and under four cycles of checkpoint inhibitor therapy. Table 3 presents all PRO and potential adverse reactions. There were no adverse reactions related to the exercise intervention.

## 3. Discussion

We investigated the effects of a 4-month exercise training program on the visuomotor, functional and cardiorespiratory capacity of a moderately active 61-year-old female patient receiving Cyberknife radiation and immunotherapy for the treatment of metastatic uveal melanoma. Present findings support the study hypothesis since EHC, functional capacity, visuomotor reaction time and cardiorespiratory fitness improved over the 4-month exercise intervention period (T0–T4). In terms of the biochemical and vascular profile, we noted a consistent stabilization of critical liver parameters throughout the exercise intervention, even in the presence of immunotherapy. Additionally, there were a few discernible alterations in the microvasculature of the tumor-affected eye before and after the intervention.

We observed decreases in visuomotor data posterior to Cyberknife treatment (T1), most likely induced by deteriorated visual impairment, as recent research suggests. Accordingly, around 50% of uveal melanoma patients treated with radiation exhibit impairments such as difficulties with steps, judging distances, walking in crowds and limited activities [4]. However, visuomotor capacity deteriorations, specifically EHC, CVRT and PVRT, cannot be explained by further impaired depth perception, as indicated by some others [19], since the low fusion capacity at baseline (T0) remained constant at T1, improved up until T3 and was then lost at T4, as the BVA data indicate. Hence, increased visual field impairment and other phenomena beyond reduced stereopsis such as binocular summation (the ratio of binocular to monocular contrast sensitivity) might have caused initial declines in EHC, CVRT and PVRT. Monocular presented stimuli require a contrast 1.4 times higher and a longer presentation time than the same stimuli presented binocularly in order to be equally detectable, as prior investigations consistently indicate [20]. In principle, observed declines are reflected in the PRO of the OPT30, showing a tripled overall visual impairment (see Table 3) and a 50% increased vision-related functional impairment score at T1. The loss of stereopsis capacity at T4 can be most likely attributed to radiation-induced tissue degradation, further deterioration of the epiretinal gliosis in the left eye, and ocular side effects of immunotherapy, although rather uncommon (~1% of patients affected) [21], or a combination of the aforementioned. Remarkably, initial declines were followed by a meaningful recovery up until T4, where all functional and most visuomotor tests exceeded baseline values (See Figure 2). Interestingly, tests that reflect enhanced motor function and did not demand dynamic visual capacity showed moderate to large improvements. Decreased dynamic visual processing, on the other hand, is well reflected in the D2 PVRT and D2 CVRT results. While the first decreased analogous due to declining visual acuity over the course of the study, the latter demonstrated a sharp decrease in motor reaction time and simultaneously a slight increase in sensory reaction time, accumulating to an overall improvement. Overall, the VFT results support, as hypothesized, that compensatory visual-coordinative training can mitigate vision-related motor function declines with respect to stereopsis and visual field. Analogous to VFT, we witnessed a similar pattern of initial decline and subsequent recovery for PA data. Precisely, sharp declines in PA levels at T1 were followed by a gradual and consistent increase, despite the presence of immunotherapy. However, it is important to note that activity levels remained below baseline (T0) values throughout the intervention period. The precise reasons for these increases are ambiguous, and it remains unknown whether they can be attributed to motivational factors induced by the intervention or are a consequence of growing restlessness or other unknown factors. Typically, visual impairment shows an inverse correlation with physical activity [22]. Accordingly, this effect is largest for visual field impairment due to the tumor location (see Figure A1), which applies to this patient.

CPET data at T0, specifically the $V{\dot{O}}_{2}$ uptake (~31 mL·kg^−1^·min^−1^) and peak power output (150 W) measurements, exceeded the typical reference values established for healthy individuals, as recently outlined by Kaminsky et al. [23], indicating an above-average physical condition at baseline. We did not observe a decline in cardiorespiratory capacity from T0 to T2 (~seven weeks apart) since the HIIT exercise regimen could be seamlessly continued after a brief recovery (4 days) from Cyberknife radiation. CPET-derived parameters, $V_{\dot{E}}$, $V{\dot{O}}_{2}$, PPO and CEPT duration, improved continuously up until T3 and slightly declined between T3 and T4.

PRO data show a largely inconsistent pattern, with overall positive developments observed in CEPT and VFT. To some extent this might be explainable by the increasing immunotherapy side-effect burden (see Table 3) and by the progressive disease status for which, thus far, no effective treatment exists in UM.

OCTA-derived parameters can be indicative of systemic adaptations due to exercise, disease and sedentary behavior, as previous investigations suggest [24]. A reduction in FAZ is typically a sign of physiological adaptation in response to exercise [25]. Although not exactly matching the regular assessment occasions in our investigation, the reductions observed in FAZ perimetry may therefore be interpreted as indicative of physiological adaptations in response to HIIT stimulus. Moreover, overall improvements in vessel density (VD) and flow density (FD) are indicators of angiogenesis processes [26]. Considering the disruptive effects of radiation on the micro vessel network and associated radiation-induced retinopathy, an improvement of OCTA values is promising, even with limitations in assessment procedure. These results align with observations from our recent investigation that show medium to large improvements in endothelial function in close proximity to the tumor in a choroidal melanoma patient following eight weeks post-radiation with HIIT [27]. However, other OCTA data are heterogeneous, and given the novelty of the approach in uveal melanoma and the limitations of our data, a definitive conclusion remains elusive.

It is highly probable that the cardiovascular deterioration evident in the CPET results at T4 results from the negative side effects of immunotherapy or the progressive tumor status on the overall physical condition. These effects are well reflected by the reported adverse effects, including faster exhaustibility and loss of power. Analogously, some PROs, like a dramatically increased HADS score, point in a similar direction. These findings align with reported adverse reactions documented in the current immunotherapy treatment protocol for uveal melanoma patients as published by Pelster et al. [14]. After a continuous increase up until T3 and a decline in T4, the overall results, even after four cycles of immunotherapy, still exceed baseline values. Perhaps, the intervention has prepared the organism and counteracted immunotherapy-induced physical deconditioning processes to a certain degree. The monitored AST and ALT blood data corroborate this trend by revealing a consistent maintenance within the established reference range during the concurrent exercise and immunotherapy period. However, whether this observation is attributable to the exercise stimulus cannot be answered due to the limitations of this study.

## 4. Conclusions

Data from this case report indicate that the 4-month exercise intervention in a patient with metastatic uveal melanoma yielded a consistent pattern of improvement in EHC, CVRT, functional capacity and CRF levels. An initial decline in most VFT and PA data was followed by a continuous recovery. For most VFT, baseline values were exceeded. Data presented here align with the substantial body of evidence in the literature on exercise oncology and add a novel comprehension of tailoring an exercise intervention for an eye tumor patient under acute treatment in order to mitigate side effects of medical treatment and contributing to recovery and overall QoL. Interestingly, our data imply that post-radiation (T1) declines in visuomotor capacity could be recovered and expanded by enhanced manual dexterity. The OCTA data results are somewhat promising. Further research is needed to understand the combined effects and underlying mechanisms of exercise, Cyberknife radiation and immunotherapy on vasculature, oxygenation and functional impairment in uveal melanoma. This rare cancer offers a unique opportunity for future investigations to elucidate exercise-induced vascular alterations in close proximity to the tumor using OCTA or comparable techniques. Theoretically, enhanced vascularization leading to reduced hypoxia could potentially improve the delivery of immunotherapy and radiotherapy treatments [28]. As response rates in the treatment of metastatic disease in uveal melanoma remain low [14,29], the potential of targeted exercise interventions should be further investigated. Future investigations need to employ a more robust methodology and approach to extend our findings to a larger cohort of uveal melanoma patients.

## Figures and Tables

**Figure 1 reports-08-00260-f001:**
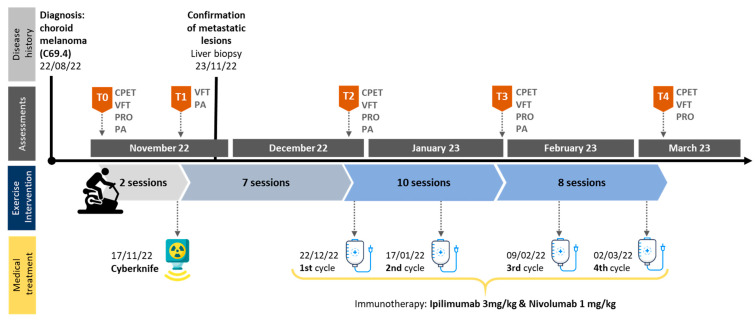
Study timeline. Abbreviations: CPET = cardio-pulmonary exercise testing; VFT = visual and functional testing; PRO = patient-reported outcomes; PA = physical activity.

**Figure 2 reports-08-00260-f002:**
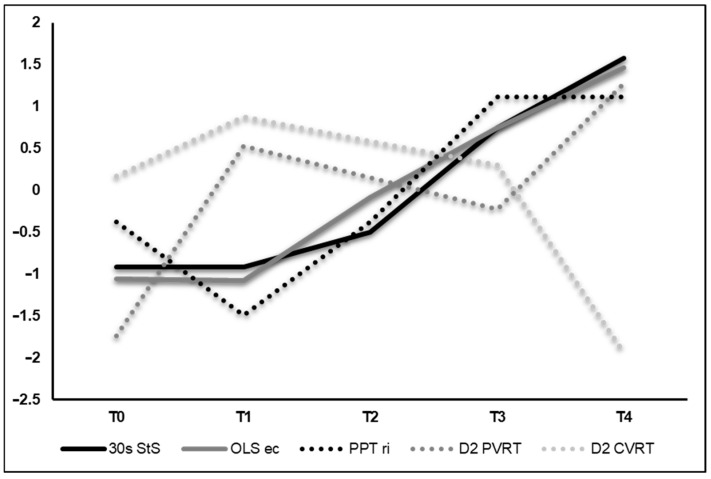
Z-transformed values of selected visual and functional tests. For 30 s StS, OLS-ec, PPT ri and D2 PVRT, an increase in the graph implies an improvement. For D2 CVRT, an increase implies a deterioration. Abbreviations: StS = sit-to-stand-test; OLS-ec = one-leg stance eyes closed; PPT ri = Purdue Pegboard Test right hand; D2 PVRT = Dynavision test peripheral vision reaction time; D2 CVRT = Dynavision test central vision reaction time.

**Table 1 reports-08-00260-t001:** Biometrical characteristics and results for cardio-pulmonary and visual-coordinative tests.

	Assessment Occasion					∆(%)			
	T0	T1	T2	T3	T4	T0–T1	T0–T2	T2–T4	T0–T4
Biometrics									
Age Sex Height (cm)	61		-	-	-				
	female		-	-	-				
	166		-	-	-				
Weight (kg) BMI (kg/m^2^)	57		56.3	56.9	56.2	-	−1.23	−0.18	**−1.40**
	20.68		20.43	20.65	20.39	-	−1.21	−0.20	**−1.40**
Tumor location Tumor size (mm)	right eye		-	-	-				
	17 × 9.7		-	-	-				
Inner ocular pressure									
right eye	18		18	16	−16	-	0.00	−11.11	**−11.11**
left eye	17		16	14	−14	-	−5.88	−12.50	**−17.65**
	Distant refractory visual acuity									
right eye	0.63		0.5	0.4	0.4	-	−20.63	−20.00	**−36.51**
left eye	0.4		0.5	0.5	0.4-	-	25.00	−20.00	**0.00**
Eye dominance	right		-	-	-				
Hand dominance	right		-	-	-				
Foot dominance	right		-	-	-				
Cardio-pulmonary exercise testing									
$V_{\dot{E}}$ (L·min^−1^)	63.11		89.59	95.36	81.98	-	41.96	−8.49	**29.90**
$V{\dot{O}}_{2}$ (L·min^−1^)	1.75		1.91	2.14	1.98	-	9.14	3.66	**13.14**
$V{\dot{C}O}_{2}$ (L·min^−1^)	2.12		2.46	2.68	2.60	-	16.04	5.69	**22.64**
RER	1.15		1.21	1.19	1.23	-	5.22	1.65	**6.96**
$V{\dot{O}}_{2\max}$ (mL·kg·min^−1^)	30.69		33.44	37.48	34.82	-	8.96	4.13	**13.46**
HR_max_	180		181	186	179	-	0.56	−1.10	**−0.56**
PPO (Watts)	150		165	180	165	-	10.00	0.00	**10.00**
Test duration (min:sec)	11:15		12:10	13:00	12:30	-	8.15	2.74	**11.11**
Hematocrit (%) [normal range: 36–45]	41		40	37	37	-	−2.44	−7.50	**−9.76**
Visual and coordinative tests									
Timed Up and Go (sec)	5.9	6.2	5.7	5.42	5.46	**5.08**	−3.39	−4.21	**−7.46**
30s sit-to-stand (rep)	17	17	18	21	23	**0.00**	5.88	27.78	**35.29**
1-Leg-Stance (eyes closed) (sec)									
best	3.82	4.26	5.31	8.44	8.92	**11.52**	39.01	67.98	**133.51**
average	3.16	3.14	4.72	6.05	7.20	**−0.63**	49.37	52.54	**127.85**
Purdue Pegboard Test (rep)									
right	16.33	15.33	16.33	17.67	17.67	−6.12	0.00	8.21	**8.21**
left	15.67	14.00	16.00	14.67	16.33	−10.66	2.11	2.06	**4.21**
both	12.67	11.67	12.67	13.00	13.00	−7.89	0.00	2.60	**2.60**
BVA									
base in break	4.00	5.00	-	10.00	unable	25.00	-	-	-
base in recovery	3.00	4.00	-	7.00	unable	33.33	-	-	-
base out break	4.00	4.00	-	21.00	unable	0.00	-	-	-
base out recovery	2.00	2.00	-	4.00	unable	0.00	-	-	-
D2: central visuomotor reaction time (sec)									
motor	0.197	0.207	-	0.138	0.126	**5.08**	-	-	**−36.04**
sensory	0.343	0.377	-	0.408	0.381	**9.91**	-	-	**11.08**
total	0.540	0.584	-	0.548	0.408	**8.15**	-	-	**−24.44**
D2: peripheral visuomotor reaction time									
hits	89	89	-	92	87	**0.00**	-	-	**−2.25**
Av.T. per hit (sec)	0.57	0.63	-	0.61	0.65	**10.53**	-	-	**14.04**

Please note: for D2 central visuomotor reaction time (sec), a decrease implies an improvement. Abbreviations: BMI = body mass index; $V_{\dot{E}}$ = pulmonary ventilation; $V{\dot{O}}_{2}$ = oxygen uptake; $V{\dot{C}O}_{2}$ = carbon dioxide production; RER = respiratory exchange ratio; PPO = peak power output; Av.T. = average time; D2 = Dynavison test.

**Table 2 reports-08-00260-t002:** Average daily physical activity data retrieved from accelerometer assessment.

	Assessment Occasions				∆(%)		
	T0	T1	T2	T3	T0–T1	T1–T3	T0–T3
Step count	9817.50	5666.20	7078.80	8216.00	**−42**	+45	**−16**
MET (>1)	1227.12	966.70	1029.25	1293.30	**−21**	+34	**+5**
MET (>1<3)	851.02	832.53	825.47	1001.43	**−2**	+20	**+18**
MET (≥3<6)	368.10	134.16	158.04	259.98	**−64**	+94	**−29**
MET (≥6)	8.00	0	45.74	31.90	**−100**	-	**+299**
TEE(kcal/d), *M(SD)*	1918.03 (1088.51)	1787.08 (771.94)	1851.34 (1041.41)	1877.09 (1003.17)	**−7**	+5	**−2**
AEE(kcal/d), *M(SD)*	632.19 (1088.51)	507.26 (771.94)	577.89 (1041.41)	598.19 (1003.17)	**−2**	+18	**−5**
Active time (min)	125.67	83.60	96.80	107.33	**−33**	−28	**−15**
Inactive time (min)	1170.20	1356.20	1335.20	1292.00	**16**	−5	**10**

The device was worn for 7 days, and calculations are based on data generated per minute. Abbreviations: MET = metabolic equivalent; TEE = total energy expenditure; AEE = activity energy expenditure; M = mean; SD = standard deviation.

**Table 3 reports-08-00260-t003:** Patient-reported outcomes and adverse reactions of immunotherapy.

		T0	T2	T3	T4
Immunotherapy-associated adverse reactions (applies: x)					
	Dry mouth	-	-	x	x
	Headache	-	-	x	-
	Rash	-	-	x	x
	Loss of power/fatigue	-	-	x	x
	Itchy skin	-	-	-	x
	Unclassifiable pain	-	-	-	x
	AST		25	23	23
	ALT		25	32	27
HADS					
	Depression	9	9	8	8
	Anxiety	13	11	11	15
	Global score	22	20	19	23
EORTC QLQ-C30 (calculated scores)					
	Physical functioning	100.00	100.00	100.00	100.00
	Role functioning	66.67	50.00	66.67	66.67
	Emotional functioning	25.00	41.67	58.33	41.67
	Cognitive functioning	66.67	50.00	66.67	66.67
	Social functioning	66.67	50.00	33.33	50.00
	Function-related QoL	65.00	58.33	65.00	65.00
	Fatigue	33.33	55.56	55.56	55.56
	Pain	66.67	66.67	33.33	66.67
	Dyspnoe	0.00	33.33	33.33	33.33
	Insomnia	66.67	66.67	33.33	66.67
	Loss of appetite	33.33	33.33	0.00	0.00
	Symptom-related QoL	81.48	71.60	84.57	80.86
	Global health status	66.67	58.33	66.67	58.33
EORTC QLQ-OPT30 (calculated scores)					
	Ocular irritation	-	44.44	55.56	50.00
	Vision impairment	11.11	33.33	33.33	33.33
	Headaches	66.67	66.67	66.67	66.67
	Worry about recurrence	-	66.67	50.00	41.67
	Problems appearance	-	66.67	50.00	50.00
	Functional problems vision	27.78	44.44	50.00	33.33
	Problems reading	66.67	100.00	100.00	100.00
	Functional problems treated eye	-	60.00	60.00	53.33
	Problems driving	50.00	66.67	50.00	83.33

Abbreviations: AST = aspartate aminotransferase; ASL = alanine aminotransferase; HADS = Hospital Anxiety and Depression Scale; EORTC = European Organization for Research and Treatment of Cancer; QLQ = quality of life questionnaire; C30 = basic module; OPT30 = complementary module for eye cancer; QoL = quality of life.

## Data Availability

The data that support the findings of this study are not publicly available as they contain information that could compromise the privacy of the research participant, but they are available upon reasonable request from MMW (corresponding author).

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
