# Peer review of "Four Months of Exercise Intervention Improved Visuomotor, Functional and Cardiorespiratory Capacity in a Patient with Metastatic Uveal Melanoma"

_reports, 2025, doi:10.3390/reports8040260_

Round 1
Reviewer 1 Report
Comments and Suggestions for Authors
– It would be helpful if the authors could provide details on the natural post-treatment history for comparison to the rehabilitation intervention. The reported improvements are otherwise difficult to assess.
– Could the authors provide any background or scientific evidence for the exercise program chosen for the patient?
– The lines 77 and 78 seem to contain extraneous contents that should be removed or replaced.
Author Response
Dear reviewer 1,
Thank you very much for your revision. We feel that the remarks have improved our mansucript's quality. Please find the responses to your remarks below.
Kind regards,
Michael Mendes Wefelnberg
Comment 1: It would be helpful if the authors could provide details on the natural post-treatment history for comparison to the rehabilitation intervention. The reported improvements are otherwise difficult to assess.
Response 1: Thank you for raising such an important remark. Obviously, we did not point out cearly enough, that, thus far, there are very limited data on this patient collective. There is one large observational study from the UK (Damato et al 2018, 2019, reference can be found the manuscript) that shows overall declining psycho-social and motor function over 10 years but a mild recovery after 1-2 years. The supoortive care needs are quite heterogeneous in this patient collctive depending on treatment (enucleation vs. radiation) and metastatic status. In our case report, the patient shows clear symptoms of depression, anxiety and limieted visual-coordinative capacity at onset of the rehabiliation intervention. Unfortunately, at this point we do not have data of a patient not participating in the rehabilitation intervention.
We have added more details on this matter in the introduction lines 48-52.
Comment 2: Could the authors provide any background or scientific evidence for the exercise program chosen for the patient?
Response 2: Thank you for this remark. Only the HIIT intervention was based on published evidence. We added the citation and reference (line 127-128) As there are no established visual-coordinative interventions for monocular deficits, we had to develop our own visual-coordinative intervention.
Comment 3: The lines 77 and 78 seem to contain extraneous contents that should be removed or replaced.
Response 3: We have deleted this. Thank you for your watchful reading.
Reviewer 2 Report
Comments and Suggestions for Authors
Is is not suffient interval for prospective case study.
Four month of exercise intervention improve visuomotor, functional and cardiorespiratory capacity in a patient with metastatic uveal melanoma is not relevant.
Author Response
Dear reviewer 1,
Thank you very much for your revision and for sharing your opinion on our manuscript. Please find the responses to your remarks below.
Kind regards,
Michael Mendes Wefelnberg
Comment 1:
Is is not suffient interval for prospective case study. Four month of exercise intervention improve visuomotor, functional and cardiorespiratory capacity in a patient with metastatic uveal melanoma is not relevant.
Response 1:
Thank you for sharing your opinion on our manuscript. As our goal was to investigate whether an exercise intervention in this quite rare patient collective is feasible and effecitve to mitigate treatment side-effects, our chosen observational period of 4 month is relevant to the collective. The severity of side-effects is highest right after treatment in these patients. Something that our interventions as it appears acutally mitigated and even improved above baseline values while visual capacity was continuously declining.
Reviewer 3 Report
Comments and Suggestions for Authors
This study addresses an important area on the role of exercise rehabilitation in a patient with metastatic uveal melanoma. Few suggestions to make the manuscript smoother and easier for reader to understand:
- Please clarify how you analyzed the data and determined improvement. Since the author mentions this is a single-patient case, formal statistical tests are not applicable (n=1). The author did not perform statistical tests, avoid using terms like "significant improvement/ significant" (paragraph 26, 263 and 276).
- Please add multiple baseline measurements or percent change comparison, describe it briefly. This allows the reader to understand whether improvement is simply observed by changes or supported by analytical rationale. Be specific, e.g. balance test improved by X%, while strength test improved by X%, instead of "7-128% improvement" (paragraph 276).
- Please clarify the patient baseline condition and treatment phase when measurement was taken (paragraph 82-114, 137-157), at which point the patient cancer treatment each assessment was conducted. This could improve the reader understand whether improvement could be due to exercise program or other treatment effects.
- Please support the OCTA findings, which are mentioned but are not well integrated into the discussion (paragraph 295-301, 386-397). Please explain the small changes are clinically meaningful or expected from exercise training?
- Suggest adding rationale for combining HIIT and visual coordinative training (paragraph 115-130), was this based on prior studies or novel hypothesis?
- Recommend organizing key outcomes in a clear table or figure. Author could utilize a table that listed patient baseline values and post-intervention values for each outcome measure (visuometer test scores, functional capacity metrics and VO2max, etc), along with the percentage change, which would benefit the manuscript. This allows the reader to appreciate the results without having to parse them from narrative.
- Follow up from comment 3. Refer to your visual and functional test data (paragraph 275-294). Some outcomes improved, others didn't (peripheral visual reaction time seem worsened). Please clarify why. Was it due to treatment side effects or progression of disease?
- Improve the current data tables & figure (table 1-3, including figure 2). Suggest the author highlighting the most important improvement in bold or using arrow to show trends and add a quick sentence summary below each table or figure to guide the reader.
- Improvise the discussion (paragraph 330-399). Please avoid suggesting that this intervention will work broadly for other patients. Do consider adjusting statement to state the results "may suggest" or "could indicate" rather than "demonstrate", this keeps the manuscript tone as evidence based.
- Referring to T4, the patient experienced decline for visual fusion capacity and VO2 max, please discuss this more directly in discussion.
- Consider adding one or two sentences regarding safety or any challenges encountered for this intervention. This allows the reader to know if the patient had any adverse events or difficulties during the 4-month exercise program.
Author Response
Comment 1: Please clarify how you analyzed the data and determined improvement. Since the author mentions this is a single-patient case, formal statistical tests are not applicable (n=1). The author did not perform statistical tests, avoid using terms like "significant improvement/ significant" (paragraph 26, 263 and 276).
Response 1: Thank you for your comment. You are absolutely right, the ters "singificant" is confusing and not related to interference statistics. Naturally, we only conducted descriptive statistical analysis. We either deleated or replaced the term with the term "meaningful" in case of iprovements or "sharp" in case of decreases.
Comment 2: Please add multiple baseline measurements or percent change comparison, describe it briefly. This allows the reader to understand whether improvement is simply observed by changes or supported by analytical rationale. Be specific, e.g. balance test improved by X%, while strength test improved by X%, instead of "7-128% improvement" (paragraph 276).
Response 2: Thank you for this important remark to improve the readability of our results. We merged table 1 and table 2 into one table as suggested and added Δ values in percent.
Comment 3: Please clarify the patient baseline condition and treatment phase when measurement was taken (paragraph 82-114, 137-157), at which point the patient cancer treatment each assessment was conducted. This could improve the reader understand whether improvement could be due to exercise program or other treatment effects.
Response 3: Thank you for the remark. We have added information on the patient's baseline conditions and treatment history in relation to the assessments (lines 96-98, 120-121, 143-145).
Comment 4: Please support the OCTA findings, which are mentioned but are not well integrated into the discussion (paragraph 295-301, 386-397). Please explain the small changes are clinically meaningful or expected from exercise training?
Response 4: Thank you very much for this remark. We have altered the discussion section to clearify the meaning of observed OCTA changes (lines 395 - 405).
Comment 5: Suggest adding rationale for combining HIIT and visual coordinative training (paragraph 115-130), was this based on prior studies or novel hypothesis?
Response 5: As we are the first internationally conducting exercise interventions in moncular deprivation and uveal melanoma, there is no published evidence. It is a complete novel hypothesis targeting the side-effects of uveal melanoma disease and treatment through exercise. The idea is the following: visual coordinative exercise to compensate for visual decline and HIIT to stimulate the microvascular network in the eye to prevent radiation retinopathy. We added and brief rationale in section 2.4 (lines 120-123).
Comment 6. Recommend organizing key outcomes in a clear table or figure. Author could utilize a table that listed patient baseline values and post-intervention values for each outcome measure (visuometer test scores, functional capacity metrics and VO2max, etc), along with the percentage change, which would benefit the manuscript. This allows the reader to appreciate the results without having to parse them from narrative.
Response 6: See response 2.
Comment 7: Follow up from comment 3. Refer to your visual and functional test data (paragraph 275-294). Some outcomes improved, others didn't (peripheral visual reaction time seem worsened). Please clarify why. Was it due to treatment side effects or progression of disease?
Response 7: She patient exhibited meaningful declines in visual acuity, coming from an already strongly limited visual capacity (see table 1), most likely due to treatment related side-effects and progression of epiretinal gliosis in the left eye. Her tumor is located at 4 o´ clock in the right eye impacting peripheral vision more than central vision. There is a quite comprehensive explanation of the divergent results in EHC and visual-motor tests. Please see lines 351 to 376.
Comment 8: Improve the current data tables & figure (table 1-3, including figure 2). Suggest the author highlighting the most important improvement in bold or using arrow to show trends and add a quick sentence summary below each table or figure to guide the reader.
Response 8: Figures 1 & 2 have been merged (see above). Most relevant change values are now highlighted in bold for Figure 1 and 2 (formerly Figure 3). Also we added a one sentence explanation to each table and figure to clear sources of potential misunderstanding. An acutal summary of findings for each table/figure would be too extensive and actually a doubling of what is written in the text.
Comment 9: Improvise the discussion (paragraph 330-399). Please avoid suggesting that this intervention will work broadly for other patients. Do consider adjusting statement to state the results "may suggest" or "could indicate" rather than "demonstrate", this keeps the manuscript tone as evidence based.
Response 9: We could not find the wordings you are refering to. The word "demonstrate" is only used once to describe the development of D2 results (line 373). There is, to our understanding, only one sentence in Conclusion section that might be misunderstood as a generalization of our findings. We altered this sentence to singular in order to avoid implications of generalization (line 429).
Comment 10: Referring to T4, the patient experienced decline for visual fusion capacity and VO2 max, please discuss this more directly in discussion.
Response 10: There is a whole paragraph discussing the stereopsis loss in T4 (357-368). Frankly, we have nothing to add to this explanation. For the decline in CEPT results, we added "the tumor progressive status" (411-4112).
Comment 11: Consider adding one or two sentences regarding safety or any challenges encountered for this intervention. This allows the reader to know if the patient had any adverse events or difficulties during the 4-month exercise program.
Response 11: Thank you for this important remark. We have added a sentence in the results section (line 332-333).
Reviewer 4 Report
Comments and Suggestions for Authors
This case report examines the effects of a four-month exercise intervention (high-intensity interval training and visual-coordination exercises) in a patient diagnosed with metastatic uveal melanoma. The study is innovative and clinically noteworthy given the limited research on rehabilitation and exercise-based supportive therapies in the field of eye oncology. Multimodal assessment methods (visuomotor testing, functional testing, CPET, OCTA, and patient-reported outcomes) are among the study's strengths.
Points requiring major correction:
The clinical significance of the findings is unclear. While the study provides percentage change values, it does not specify whether these changes are clinically significant. Without information such as effect size, variability in measurements, or a minimal clinical significance threshold, it is not possible for the reader to assess the clinical significance of the observed improvement.
The interpretation of the OCTA results is overly causal. A direct relationship is being established between exercise and the observed vascular changes. At the case report level, asserting that exercise causes microvascular changes when there are concurrent effects such as radiotherapy and immunotherapy is a matter of scientific caution. In this section, probability language should be used instead of expressing definitive results.
The possibility of a learning effect in visuomotor and functional tests has been ignored. Tests such as the Purdue Pegboard, Dynavision, and Timed-Up-and-Go may show improvement with repetition. It is not possible to distinguish between the test-relearning effect and improvements resulting from the exercise intervention. The authors should discuss this limitation.
The clinical explanation for changes in visual performance is unclear. Stereopsis measurements with BVA initially improved and then disappeared completely. The relationship of this change to radiation retinopathy, immunotherapy side effects, or disease progression should be discussed, and it should be clearly stated that it is not related to exercise.
There is inconsistency between patient-reported outcomes (quality of life questionnaires) and objective measurements. Despite improvements in functional and performance tests, there is a deterioration in some quality of life scores. This contradiction should be discussed in the context of immunotherapy side effects and the patient's psychological burden.
The introduction section appears to lack reference to recent molecular studies related to the metastatic process in uveal melanoma. The following study should be included: doi:10.1177/03936155221088886
Author Response
Comment 1: The clinical significance of the findings is unclear. While the study provides percentage change values, it does not specify whether these changes are clinically significant. Without information such as effect size, variability in measurements, or a minimal clinical significance threshold, it is not possible for the reader to assess the clinical significance of the observed improvement.
Response 1: Thank you for this important remark. However, to our knowledge, this is the second paper (the first was also published by our working group) internationally that deals with a supportive care approach for uveal melanoma. There is no experience with this patient group in this regard. How motor improvements relate to clinically relevant effects has not been examined yet but is something that we are currently focussing on in a larger randomised-controlled interventional study. The primary aim of our case study was to assess feasibility of an exercise intervention for uveal melanoma. As this is a case study effect sizes could not be calculated.
Comment 2: The interpretation of the OCTA results is overly causal. A direct relationship is being established between exercise and the observed vascular changes. At the case report level, asserting that exercise causes microvascular changes when there are concurrent effects such as radiotherapy and immunotherapy is a matter of scientific caution. In this section, probability language should be used instead of expressing definitive results.
Response 2: We agree. However, reviewer 3 demanded more insights into the potential effects of exercise on the OCTA results. So we are in minor conflict here, who to satisfy. Maybe the editor can decide?
Comment 3: The possibility of a learning effect in visuomotor and functional tests has been ignored. Tests such as the Purdue Pegboard, Dynavision, and Timed-Up-and-Go may show improvement with repetition. It is not possible to distinguish between the test-relearning effect and improvements resulting from the exercise intervention. The authors should discuss this limitation.
Response 3: We agree. This is the case for any motor task that involves at least some elements of coordination. There is extensive research on the PPT (the test exists since the 1940s) in various contexts and under various conditions. Test-retest and interrater-reliability are very high. The D2 is a more novel approach but shows high robustness as well. We added a sentence to inform the reader that a learning effect can exist. Yet we dought that learning effects explain the partly divergent results we see in this case report.
Comment 4: The clinical explanation for changes in visual performance is unclear. Stereopsis measurements with BVA initially improved and then disappeared completely. The relationship of this change to radiation retinopathy, immunotherapy side effects, or disease progression should be discussed, and it should be clearly stated that it is not related to exercise.
Response 4: We are providing a clinical explanation for this observation in line 365-368. The patient started with very low fusion capacity and hase a tumor disease in one eye and an epiretinal gliosis (which is in nature progessive) in the other. Most likely, this is the reason for the loss of stereopsis. To clearify more, we added the word "severe" in line 94 in the methods section. As there is no evidence of any sort that exercise can cause visual impairments we feel it is absolutely unnessecary to state the obvious and that such a sentence rather confuses the reader. Investigation on ocular health rather show beneficial effects of exercise on ocular physiology and function. So we would like to exclude this suggestion from our paper.
Comment 5: There is inconsistency between patient-reported outcomes (quality of life questionnaires) and objective measurements. Despite improvements in functional and performance tests, there is a deterioration in some quality of life scores. This contradiction should be discussed in the context of immunotherapy side effects and the patient's psychological burden.
The introduction section appears to lack reference to recent molecular studies related to the metastatic process in uveal melanoma. The following study should be included: doi:10.1177/03936155221088886
Response 5: Thank you for this remark. We have added two sentences to stress the inconsistency in PRO and VFT. Thank you for providing the reference. Has been added to the paper.
Round 2
Reviewer 1 Report
Comments and Suggestions for Authors
Thank you for the response. I have no further questions.
Reviewer 3 Report
Comments and Suggestions for Authors
Thank you for your careful revisions. I have reviewed the updated manuscript, and I am pleased to see that all previously raised comment and have adequately addressed. The clarifications, additional details, and improved presentation have strengthened the manuscript considerably.
Reviewer 4 Report
Comments and Suggestions for Authors
Thanks for the implementation of the suggestions.